# Computational Analysis of African Swine Fever Virus Protein Space for the Design of an Epitope-Based Vaccine Ensemble

**DOI:** 10.3390/pathogens9121078

**Published:** 2020-12-21

**Authors:** Albert Ros-Lucas, Florencia Correa-Fiz, Laia Bosch-Camós, Fernando Rodriguez, Julio Alonso-Padilla

**Affiliations:** 1Barcelona Institute for Global Health (ISGlobal), Hospital Clínic—University of Barcelona, 08036 Barcelona, Spain; albert.ros@isglobal.org; 2Animal Health Research Centre (CReSA, IRTA-UAB), IRTA, Bellaterra, 08193 Barcelona, Spain; flor.correa@irta.cat (F.C.-F.); laia.bosch@irta.cat (L.B.-C.); 3OIE Collaborating Centre for the Research and Control of Emerging and Re-emerging Swine Diseases in Europe (IRTA-CReSA), Bellaterra, 08193 Barcelona, Spain

**Keywords:** African swine fever virus, ASFV, swine, pigs, B cell, CD4^+^ T cells, CD8^+^ T cells, epitopes, vaccine ensemble

## Abstract

African swine fever virus is the etiological agent of African swine fever, a transmissible severe hemorrhagic disease that affects pigs, causing massive economic losses. There is neither a treatment nor a vaccine available, and the only method to control its spread is by extensive culling of pigs. So far, classical vaccine development approaches have not yielded sufficiently good results in terms of concomitant safety and efficacy. Nowadays, thanks to advances in genomic and proteomic techniques, a reverse vaccinology strategy can be explored to design alternative vaccine formulations. In this study, ASFV protein sequences were analyzed using an in-house pipeline based on publicly available immunoinformatic tools to identify epitopes of interest for a prospective vaccine ensemble. These included experimentally validated sequences from the Immune Epitope Database, as well as de novo predicted sequences. Experimentally validated and predicted epitopes were prioritized following a series of criteria that included evolutionary conservation, presence in the virulent and currently circulating variant Georgia 2007/1, and lack of identity to either the pig proteome or putative proteins from pig gut microbiota. Following this strategy, 29 B-cell, 14 CD4^+^ T-cell and 6 CD8^+^ T-cell epitopes were selected, which represent a starting point to investigating the protective capacity of ASFV epitope-based vaccines.

## 1. Introduction

African swine fever virus (ASFV), the causative agent of African swine fever (ASF), stands among the most lethal pathogens that infect domestic pigs (*Sus scrofa*) worldwide. First described in the 1920s in Kenya after the introduction of European domestic pigs in the country [1], disease outbreaks were mostly circumscribed to eastern and southern sub-Saharan Africa during the first half of the 20th century. Since then, it has spread across the globe causing epidemics not just in other African countries, but also in Europe, the Caribbean, South America, Oceania and Asia. The last importation of ASFV from Africa to Europe triggered the biggest ASF epidemic ever documented, becoming the number one threat for today’s global swine industry [2,3]. Due to its potential devastating impact, ASF is listed as a disease of obliged notification to the World Organization for Animal Health (OIE).

In Africa, the main ASFV reservoirs are African wild pigs such as warthogs (*Phacochoerus africanus*) and bushpigs (*Potamochoerus larvatus*), where asymptomatic infections occur. In these areas, maintenance of the ASFV infectious cycle involves tick vectors of the genus *Ornithodoros*, which are responsible of the transmission to domestic pigs [4]. The main risk factor for today’s transmission of the virus in Europe and Asia is the direct contact with infected material such as carcasses or pork products, as well as transmission through contact with infected individuals [5]. Wild boars in Eurasia are also susceptible to the disease, showing similar symptoms and mortality rates as domestic pigs, hampering efforts to control ASF in the continent [6]. ASF causes an acute hemorrhagic fever that is almost 100% fatal with the most virulent variants. There is no vaccine or treatment available, and the only effective way to control the spread of the virus is by sacrificing all pigs in the affected area, i.e., those infected and those that might have been in contact with them. Safe disposal of carcasses and other contaminated materials is required, which involves implementing maximum control and biosafety measures, as well as imposing restrictions to the movements of pigs and of their derived products [6].

ASFV, the only member of the family *Asfarviridae*, is a large (~200 nm) virus belonging to the nucleocytoplasmic large DNA viruses (NCLDVs) group [7]. It has a complex multi-layered architecture, consisting of several lipid and protein layers (Figure 1). Its genome is encapsulated by a first protein layer consisting of a core shell inner capsid, which in turn is covered by an inner lipid bilayer envelope, itself contained in an icosahedral protein capsid [7,8]. Finally, extracellular particles gain an external lipid envelope when they egress through the plasma membrane [9]. The virus’ main target cells are macrophages, but it can also infect dendritic cells and other monocytes [10]. The viral genome consists of a single double-stranded DNA molecule between 170 and 193 Kbp long that encodes over 150 open reading frames (ORFs) including its own replication machinery, gene regulation, host defense evasion factors, and its whole set of structural proteins. Differences in genome length and number of ORFs between viruses are mostly attributed to different numbers of multigene family (MGF) genes located in variable regions at both ends of the genome [11], while more subtle differences can be found in genes of the central conserved region [12,13]. Notably, the functions of a sizable portion of the virus proteins are still unknown [14].

Several variants of the virus from different countries, outbreaks and with varying degrees of virulence have been identified. The first complete genome sequenced was from an isolate from Badajoz (Spain) in the early 1970s, adapted to grow in Vero cells (Ba71V) [15]. After that, many other genomes have become available, and as of August 2020, 46 proteomes from different variants or isolates are accessible at UniProt [16]. These can be categorized in 22 different genotypes, established upon analysis of the 3′-end sequence of the major capsid protein p72 gene (B646L) [17]. Geographically, the largest genotype variability is found in Africa, while in Europe, America and Asia only isolates from genotypes I and II have been found [18]. So far, there is no evidence correlating genotype and virulence [19].

Pigs surviving the disease develop immunity, and inoculation of attenuated viruses induces immune responses and variant-specific protection [20,21,22,23,24]. Therefore, a vaccination strategy could be pursued, albeit safety issues have hampered field implementation of live attenuated vaccines. Other immunization approaches like protein subunits or DNA constructs were not able to mimic the protection levels achieved by live attenuated viruses, despite some of them inducing specific responses [25,26,27,28,29,30]. Nonetheless, the few in vivo protection data available, using both live attenuated viruses and subunit vaccines, have allowed identifying molecular and immunological mechanisms involved in such protection. On the one hand, it has been widely established that specific T-cells play a key role [20,26,31,32,33], although the specific T-cell determinants triggering protective responses remain largely unknown. On the other hand, several ASFV serological determinants have been described [34,35,36], yet their protective role is unclear. Nonetheless, passive transfer of ASFV-specific antibodies was reported to confer protection [37,38], evidencing their role in controlling the infection. Additionally, new evidence suggests that microbiota might play an important part in the defense against the virus, as fecal transplants from warthogs to pigs show that partial protection against ASFV is achieved [39].

An alternative vaccine design strategy targeting multiple antigens, capable of inducing protective responses without compromising safety could be an approach to address the virus biological complexity. In this regard, the application of reverse vaccinology (RV) can be very helpful for the screening of the growing number of ASFV sequences in search of the best target epitopes. Devised to develop vaccines for pathogens difficult to isolate and culture, RV is based on the computational analysis of the pathogens’ genomic and/or proteomic data to identify the proteins of highest interest for a hypothetical vaccine [40,41]. The use of an in-silico approach also permits to consider viral variability, specifically identifying invariant regions, helpful for the design of pan-protecting vaccines [42]. Similarly, it could serve to avoid sequences not necessary to induce protective responses or that might lead to exacerbation of the disease [33,43,44]. Inspiringly, there is already an approved vaccine produced by RV against Neisseria meningitidis serogroup B licensed in 2013 [45].

## 2. Results

### 2.1. Analysis of ASFV Protein Space

A total of 7088 ASFV protein sequences were downloaded from the 46 available proteomes of the virus in UniProt (Appendix A). In total, seven proteomes were discarded due to their aberrant number of proteins: Kyev 2016 (UP000321214), Poland 2016 o23 (UP000274966), Poland 2016 o9 (UP000268777), Poland 2017 C220 (UP000282187), Poland 2016 o10 (UP000278405), Sardinia Sassari 2008 (UP000266411) and South Africa 1985/SPEC 57 (UP000423628).

Protein clustering with CD-HIT [46] produced 459 clusters. In order to consider ASFV variability in each resulting cluster, we filtered out clusters with sequences that originated from fewer than 14 distinct proteomes. This threshold was selected given this is the number of available ASFV proteomes from genotype II (Georgia 2007/1), in an attempt to keep all possible ORFs belonging to this genotype. Thus, we considered for further analysis the 173 clusters that contained proteins from 14 or more different ASFV proteomes. Remarkably, calculations of Shannon entropy [47] carried out to quantify position variability showed that 82.5% of all amino acid positions among the clusters’ multiple sequence alignments (MSAs) were totally invariant, i.e., had an entropy *H* = 0 (data not shown). Thus, in order to be as stringent as possible, all subsequent analysis was done with the invariable consensus proteome derived from assigning a Shannon entropy threshold of *H* = 0.

### 2.2. Identification of ASFV Putative-Exposed Proteins

So far only one protein has been described on the surface of the ASFV virion: the CD2 homolog (ORF: E402R) [14]. An extra effort was made with the aim to identify viral proteins with the potential to be found on the outer surface of the membrane of infected cells and/or the extracellular virion and therefore susceptible for antibody neutralization. To identify putative-exposed proteins, we ran subcellular localization prediction tools over the Georgia 2007/1 proteome (UP000141072), selecting a total of 49 putative-exposed proteins based on criteria of annotation, and the predicted presence of a signal peptide and/or at least one transmembrane domain. Putative-exposed proteins included those predicted to be exposed, as well as those previously described as forming part of the external capsid, potential targets of antibodies in the case of non-enveloped infectious particles (see Appendix A). All of them were tagged as “exposed” and clustered with the rest of ASFV proteomes to find all ASFV “exposed” homologues. Amongst the 173 clusters that included 14 or more ASFV proteomes, only 43 had “exposed” proteins and were selected for further exploration of B-cell epitopes presence.

### 2.3. Selection of B-Cell Epitopes

We found 102 experimentally validated and unique epitopes of the virus at the Immune Epitope Database [48], 76 of which were B-cell epitopes. We kept those that originated from putative-exposed proteins, and then parsed their full sequence against the ASFV invariant “exposed” proteome. As a result, we found nine conserved B-cell epitopes. To further curate this selection, we compared these sequences with *Sus scrofa*’s proteome and its microbiota using a BLASTP search, as described in Material and Methods. We discarded those sequences that had an identity >70% against *Sus scrofa* proteins or those within its microbiota, reducing the list to four conserved B-cell epitopes from three different proteins: two were from the major capsid protein p72 (B646L), and two from the inner envelope proteins p22 (KP177R) and p54 (E183L), respectively (Table 1). Epitopes VCKVDKDCGSGEHCV (p22) and FPENSHNIQTAGKQD (p72) were found to share residues with sequence-based predicted epitopes; the latter also shared residues with a structure-based predicted epitope.

### 2.4. De Novo Predicted B-Cell Epitopes

In addition to the IEDB legacy B-cell epitopes, we pursued two methodologies to predict new ones: A sequence-based and a structure-based approach. Following the sequence-based approach, we retrieved a list of 42 potential B-cell epitopes conserved among ASF viruses. For those epitopes from proteins where we had predicted transmembrane domains, we mapped them to assess their positioning as being “outside”. Any epitopes mapping to the inner side of transmembrane proteins or within transmembrane locations were discarded. In addition, we visualized potential epitopes in the major capsid protein p72 (PDB ID: 6L2T) with PyMOL, and discarded those that were not exposed, i.e., facing outwards the virion outer capsid. By doing so we reduced the list to 29 epitopes, and deselected two additional ones since they had an identity >70% against protein sequences in *S. scrofa* or its microbiota. The selected 27 predicted B-cell epitopes originated from 13 different proteins (Table 2). Some of these epitopes share residues with sequences validated at the IEDB not necessarily selected in Table 1. This is due to the fact that epitopes from IEDB were parsed in their entirety against the consensus proteome, which caused the discard of long sequences with variable positions and/or sequences that showed an identity >70% to proteins in pigs.

For the structure-based approach, we applied specific constrains to flexibility and relative solvent accessibility (RSA) of residues on the two structures of ASFV major capsid protein p72 available in PDB [49]: 6KU9 [50] and 6L2T [9]. We identified 13 potential epitopes that were also conserved in the “exposed” invariant proteome. Upon discarding non-unique peptides, consequence of the p72 homo-trimeric assembly onto capsomers and the fusion of overlapping epitopes, we ended up with eight unique structure-predicted B-cell epitopes. Just three of them had an identity ≤70% to *S. scrofa* proteins and those of its microbiota, all from PDB structure 6L2T. Upon their visualization in PyMOL to inspect accessibility, we discarded one located to the base of the protein and thus non-exposed [7,9]. The remaining two potential epitopes are shown in Table 3 and Figure 2. Both these epitopes were also predicted in the sequence-based approach (see Table 2).

### 2.5. Selection of CD4^+^ T-Cell Epitopes

Out of the 102 experimentally validated epitopes from IEDB, 26 were T-cell epitopes. These lacked the information regarding MHC binding, but due to their length (≥15 amino acids) we considered them as CD4^+^ T-cell epitopes. Only three were conserved, all from the same protein CD2 homolog. However, none of these had an identity ≤70% in the BLASTP searches against proteins in pigs and their microbiota, so they were discarded.

### 2.6. De Novo Predicted CD4^+^ T-Cell Epitopes

We did not find ASFV CD4^+^ T-cell epitopes at IEDB that complied with the selection criteria, so we predicted new CD4^+^ T-cell epitopes for their potential inclusion in the vaccine ensemble. Applying IEDB MHC class II binding prediction tool [51] on the invariant proteome produced a total of 13,342 potential 15-mer epitopes, and we carried on with the 273 sequences within the top 0.1% binders; most of them 15-mer segments of larger peptides. Fusion of overlapping sequences yielded 74 unique peptide sequences. We discarded any epitope predicted to bind with only one allele and hence, we kept 16 sequences. Upon BLASTP searches against proteins in *S. scrofa* and its microbiota we selected 14 potential epitopes from 13 different proteins that had an identity ≤ 70% to those (Table 4).

### 2.7. Selection of CD8^+^ T-Cell Epitopes

No experimentally validated CD8^+^ T-cell epitopes were available at IEDB, and a literature search did not find any epitopes present in the invariable proteome. Thus, we predicted new CD8^+^ T-cell epitopes using a sequence of steps that mimicked the natural processing pathway of these epitopes within the cell [52].

Proteasomal processing of the invariable proteome generated 277 conserved peptides that had a minimum length of nine amino acids; any fragment shorter than nine amino acids was discarded. We created all possible overlapping 9-mer combinations of peptides from those proteasomal-derived fragments that were larger than nine amino acids long, which produced 669 potential epitopes. We applied a log(IC_50_) value of 1 as threshold for TAP [53], which restricted the list of potential epitopes to 40 peptide sequences. IEDB MHC class I binding tool predicted 33 of them within the top 1 percentile [54,55]. Of these, 22 had an identity to *S. scrofa* proteins ≤70%. However, regarding the potential cross-reactivity to proteins in the microbiota, we increased an identity threshold of ≤80%. Final list of selected potential CD8^+^ T-cell epitopes contained only six peptides from six different proteins is shown in Table 5.

## 3. Discussion

Since 2018, dozens of countries in Africa, Europe, Asia and Oceania have reported cases of ASF. Official figures of total animal losses in Asia since the first outbreak in China in 2018 are estimated to be over 6 million [56]. However, the real number will probably be much higher. While the UN Food and Agriculture Organization (FAO) official toll in China one year after the first outbreak was around 1.2 million dead pigs [57], other estimates pointed to losses of at least 40% of China’s 360 million pig population during the same period of time [58,59].

With no treatment or vaccine at hand, figures above illustrate that there is an obvious and urgent need for therapeutics against ASFV, but no success in developing a safe and effective vaccine has yet been achieved [30]. Presently, most advanced immunization efforts have been obtained using live attenuated viruses, either naturally occurring or generated in the lab [30]. For example, immunization with the low virulent variant NHV/1968 provided protection against the closely related, but highly virulent, L60 [60]. Protection was thus only partially attained against a heterologous challenge and harmful side-effects were seen in the inoculated pigs [60]. Another example of a promising vaccination strategy has been the use of the Ba71 variant deficient in the protein CD2 homolog [20]. This attenuated virus offered protection against the parental virus, and also against challenge with heterologous variants E75 and Georgia 2007/1, without causing significant side-effects [20]. Strikingly, the same principle failed to offer protection when using a CD2 homolog-deficient Georgia 2007/1 virus [61], highlighting the phenotypical diversity of ASF viruses. A plausible explanation for this might be the existence of redundant ORFs that can cover essential functions of the CD2 homolog. Indeed, Georgia 2007/1 shows greater complexity compared with Ba71, the former having a larger number of ORFs in comparison with the later [13]. This further reinforces our focus on Georgia 2007/1, since we want to tackle this complexity as much as possible.

Vaccination with genetically modified viruses has also been attempted with varying results in terms of the evoked levels of immunization, acquisition of protection against a heterologous challenge, or advent of harmful side-effects [30]. Immunization with ASFV subunits of specific viral proteins has been pursued as well. This type of vaccine is safer than aforementioned approaches and there are various examples of already licensed products based on subunits immunization [62]. However, for ASFV this strategy has only offered partial or no protection at all [63].

In this work, we have used a computer-assisted methodology to identify a group of ASFV epitopes, which could be of interest for the design of an epitope-based vaccine against ASF (Figure 3). With the scope of evoking a wide and robust immune response able to protect against heterologous challenges, we have prioritized potential epitope sequences that are strictly conserved among the different ASFV isolates. Selecting conserved epitopes also has the advantage of hypothetically targeting highly conserved regions, which might be biologically essential for the virus and thus good vaccine targets [42,64]. However, any chosen epitope must be able to awake a truly protective response, as chronic infection involving non-neutralized ASFV has been widely described, even in the presence of neutralizing antibodies [65].

Several strategies were followed to select B-cell, CD4^+^ and CD8^+^ T-cell epitopes, including a legacy experimentation approach interrogating the IEDB repository [66], and de novo prediction of epitopes by means of well-established immunoinformatics tools [51,67]. Sequences were prioritized on the basis of an identity below a 70% threshold to proteins in *S. scrofa* and its microbiota. The rationale for this is that, firstly, any sequence dissimilar to those in the host will have a higher chance of being immunogenic upon delivery [68], indeed a desirable feature in a prospective vaccine. Secondly, for safety reasons, as sequences that are dissimilar to those in the host should avoid cross-reactive responses. Additionally, since microbiota has recently been confirmed as a relevant player in ASF resistance [39], any optimal vaccine should be able to induce both a systemic and a mucosal immunity, capable to block the pathogens at the entry site, while avoiding an undesired vaccine-induced response against the host microbiota. The fact that a ≤80% threshold was used in the microbiota BLASTP during the prioritization of CD8^+^ T-cell epitopes was associated to the short length inherent to these (9-mers), which would otherwise be discarded at stricter thresholds. It must be highlighted that due to the lack of species-level data for the pigs’ microbiota, BLASTP was run against all species within each of the genera reported to be present in pigs, which might include species not truly present in them. Thus, a ≤80% threshold was decided as good trade-off between low identity and BLASTP hits.

Antibodies are relevant in ASFV immunity as the transfer of anti-ASFV immunoglobulin G (IgG) from surviving pigs can protect against infection [38]. We followed three different approaches to select B-cell epitopes that might be of interest: one legacy experimentation approach, searching the IEDB for conserved and experimentally validated epitopes (Table 1), and two de novo prediction methods, with a sequence-based approach based on BepiPred-2.0 [69] (Table 2), and a structure-based approach using 3D structures from PDB [49,67], only applicable in the case of p72 (Table 3).

ASFV is an enveloped virus, and thus the capsid is hidden under a lipid bilayer (Figure 1). To date, the only protein known to be embedded in the outer lipid envelope is the CD2 homolog protein (E402R) [14,70], for which we predicted an epitope following the sequence-based approach. This protein is one of the most divergent among different ASF viruses [71], so the discovery of a conserved epitope between different isolates is relevant. While this is an enveloped virus, it has been reported that both the extracellular and the intracellular, “naked” particles, are infectious [72]. ASFV infection can cause cell lysis, and released intracellular particles can be already infectious [73]. These “naked” viruses would have an antigen surface mainly consisting of copies of the major capsid protein p72 (B646L), which constitutes the majority of the outer capsid. In addition, the existence of neutralizing antibodies against this protein has been described [74], justifying the prioritization of B-cell epitopes from p72. Our search found two conserved sequences from the IEDB, in addition to two epitopes predicted with the structure-based approach and seven with our sequence-based approach. Remarkably, both epitopes identified from the analysis of p72 3D structures (Figure 2) contain residues which match sequences predicted by the sequence-based approach (Table 2) and experimentally validated sequences from IEDB, one of which is also found within the conserved sequence FPENSHNIQTAGKQD (Table 1). In addition to p72, another protein for which we have predicted epitopes is M1249L, which Wang et al. [9] described as an essential part of the skeleton of the capsid, and thus may be targeted by antibodies in a “naked” particle scenario.

Other selected potential B-cell epitopes originate from proteins below the capsid, embedded in the inner lipid layer. Epitopes for proteins p54 (E183L) and p22 (KP177R) have both been found with the IEDB legacy experimentation search and the sequence-based approach, and in the particular case of p22, the sequence-based predicted epitope shares more than half of its sequence with the one experimentally validated. In addition, we predicted one epitope in protein p17 (D117L) and five in p49 (B438L), the later having one epitope with residues found in sequences validated at IEDB. As mentioned above, all these proteins are located within inner layers of the viral particle, and the protective role of antibodies against them may not appear clear. Nonetheless, neutralization mediated by antibodies against inner-particle proteins p22, p54 and p30 (CP204L) has been described [25,74], suggesting the existence of a mechanism through which these proteins would be accessible to antibodies. It may be hypothesized that the virion layers are dynamic structures rather than static ones, and thus proteins present in them could be susceptible to neutralization. In addition, epitopes from proteins B169L, C257L, B475L, E146L, trans-prenyltransferase B318L and MGF 110-9L were also predicted, but verification of their location in the viral particle could not be confirmed through literature [14]. These proteins were selected in our effort to identify viral proteins which have the potential to be putatively exposed.

All this said, it is important to highlight that previous vaccination studies have described that some of these antigens may cause exacerbation of the disease, presumably by an antibody-dependent enhancement of the infection, as it was the case with p30, p54, p17, p72 and even the CD2 homolog [33,43,44]. Extra caution should be taken in their experimental validation. In addition, a limitation of the predictive approach is that with the methodology followed we cannot predict conformational B-cell epitopes. Non-protein epitopes like glycan-based ones were not retrieved either.

Previous experiments with live-attenuated vaccines [31] and DNA vaccines [26] have shown that CD8^+^ T-cells play a major role in developing immunity against ASFV, but we did not find any conserved experimentally validated CD8^+^ T-cell epitopes. We had to entirely rely on computational methods to predict them. The six epitopes ultimately selected were predicted to bind to 38 out of the 45 SLA class I alleles available in the prediction tool [51]. Binding to that variety of SLA alleles would ensure wide population coverage. In fact, just with epitopes MAMQKLFTY and KRHENIWML, the maximum possible coverage would be attained (Appendix A). Notably, all selected CD8^+^ T-cell potential epitopes originate from the replication and transcription machinery of the virus: putative DNA-directed RNA polymerase subunit 5 homolog (D205R), uncharacterized protein D339L (described as an RNA polymerase subunit elsewhere [14]), DNA-directed RNA polymerase subunit beta (EP1242L), ribonucleoside-diphosphate reductase (F334L), DNA topoisomerase 2 (P1192R), and uncharacterized protein G1340L (described elsewhere as a probable early transcription factor [14]). A series of CD4^+^ T-cell epitopes were also selected. Naive T_h_ cells are typically activated by antigen-presenting cells (APCs), which are coincidentally the main target of ASFV (macrophages, monocytes, and dendritic cells) [10]. This phenomenon could hamper APCs ability to present antigens, an essential step to develop an efficient adaptive immune response [75]. CD4^+^ T-cells play a fundamental role both in the balance and the maintenance of cytotoxic and humoral responses, so any vaccination efforts should consider the role of these cells. In this study, we did not find any experimentally validated CD4^+^ T-cell conserved epitope at IEDB, so we were forced to predict them. Our de novo approach found 14 potential CD4^+^ T-cell epitopes (Table 4). Remarkably, both CD2 homolog and p72, drivers of a B-cell mediated response, provide one potential CD4^+^ T-cell epitope each. It must be noted that most immunoinformatics tools have been designed and calibrated with data from a human immunological context, which might have an influence on the prediction of ASFV CD4^+^ T-cell epitopes.

Ultimately, we prioritized 6 CD8^+^ T-cell epitopes, 14 CD4^+^ T-cell epitopes and 29 B-cell epitopes that are unique and conserved. This list offers a wide array of possibilities for a possible ASFV epitope-based vaccine, taking into consideration variability between all viral variants described to date. Next steps will include the experimental validation of those epitopes coming from a predictive approach, and the overall assessment of their individual and/or group capacity to evoke an immune response that is protective against viral challenge. Upon elucidating this, they may well serve to design a safe and efficacious epitope-based vaccine ensemble for ASF.

## 4. Materials and Methods

### 4.1. Collection of ASFV Proteomes

All available proteomes with their respective proteins were downloaded from UniProt [16] on 25 August 2020. All sequences were correspondingly labeled to identify their proteome of origin and a priority value to be used in the multiple sequence alignment step. Sequences derived from Georgia 2007/1 (UP000141072), which is responsible of the current epidemic in Eurasia, were labeled with higher priority.

### 4.2. Collection of Experimentally Validated ASFV Epitopes

Experimentally validated epitopes of ASFV were downloaded from the Immune Epitope Database (IEDB) [48] on August 25th, 2020. Only epitopes derived from positive assays were collected, as either B-cell or T-cell epitopes.

### 4.3. Clustering of ASFV Proteins and Generation of Multiple Sequence Alignments

We used CD-HIT [46] to reduce the redundancy of the ASFV protein sequences retrieved. We ran the program using an 80% identity cutoff and a minimum of 75% length coverage to avoid clustering of fragmented proteins present in some proteomes. Then, sequences within the remaining clusters were aligned using MUSCLE [76] under default settings. As a result, we obtained an MSA file for each cluster.

### 4.4. Generating an ASFV Invariable Proteome

We used the Shannon entropy calculation to quantify the sequence variability in the MSA files [47,77]. We generated a preliminary consensus sequence for each MSA, product of the most common amino acid in each position. However, any position with entropy (*H*) higher than the selected threshold was masked with an asterisk. Since this consensus may not reflect a sequence present in nature, we determined that the true consensus would be the ASFV sequence of each MSA most similar to the masked consensus, with the corresponding masked positions according to the entropy threshold. Sequences from Georgia 2007/1 had preference when the resemblance was the same between different viruses. Nevertheless, since we established a strict threshold (*H* = 0; i.e., no variability) all sequences shared the same resemblance to the preliminary masked consensus, which results in Georgia 2007/1 being the representative in all cases. We finally generated a FASTA file containing all the masked consensus sequences from all the different MSA, which we considered as the invariable proteome [78].

### 4.5. Prediction of CD8^+^ T-Cell Epitopes

We used NetChop to predict mammalian proteasome processing [79]. In brief, NetChop was fed with the masked consensus sequences and ran using the C-term 3.0 method and a 0.5 threshold. We parsed its results to yield a file with all peptide fragments product of the predicted proteasome degradation. Since the majority of peptides presented by MHC class I are 9-mers, we discarded all fragments shorter than nine amino acids long, while fragments longer than nine amino acids were subdivided into all possible overlapping 9-mer combinations. Since the transporter associated with antigen processing (TAP) binding is a needed stage for most MHC-I-binding peptides, we used an online server that applied the model described by Besser and Louzoun [80] to estimate the TAP binding affinity of the predicted peptides. Finally, we used an IEDB MHC I binding prediction tool to predict which peptides would bind more effectively to pig MHC I complex [51]. We used all 45 available Swine Leukocyte Antigen (SLA) class I alleles, and ran the “IEDB-recommended” method for each allele.

### 4.6. Prediction of CD4^+^ T-Cell Epitopes

For this step we used IEDB MHC class II binding prediction tool with default values and “IEDB-recommended” method for each allele [51] on the ASFV invariant proteome. Given that default epitope length of this tool is 15 aa, we first discarded all those peptide fragments in the invariant proteomes with a length inferior to that, whereas all those fragments > 15 residues long were divided into their 15-mer combinations by the tool itself. Since there are no available SLA class II alleles for this tool, we used the HLA class II reference set consisting of 27 alleles available at IEDB [81]. It has been described that HLA-II can be used to approximate predictions for other mammals [82].

### 4.7. Prediction of B-Cell Epitopes

B-cell epitopes drive antibody-mediated responses and they usually locate to surface exposed moieties of the antigens. Thus, we first selected ASFV candidate proteins that were either described or predicted to be exposed. For the latter we used an array of computational tools on the Georgia 2007/1 proteome, consisting of TMHMM, SignalP and TargetP to respectively predict the presence of transmembrane domains, signal peptides and the subcellular localization of protein sequences [83]. We additionally used DeepLoc, which predicts the subcellular localization of eukaryotic proteins [84], and we also checked for the presence of glycophosphatidylinositol anchor signals using the PredGPI server [85]. We ranked proteins according to results given by these tools with special attention to proteins with transmembrane domains, which would indicate membrane-bound proteins, and thus having a higher probability of being exposed. Two strategies were then followed to predict B-cell epitopes: (i) a sequence-based, and (ii) a structure-based approach.

(i) For the sequence-based approach, all proteins expected to be “exposed” were tagged as such and incorporated to the proteome data files. Then, CD-HIT was used and any resulting clusters that did not have at least one “exposed” sequence were discarded. Generation of the “exposed” consensus invariant proteome was made following the same steps as described above. Then, we ran BepiPred-2.0 on the unmasked consensus proteome file, with a default threshold of 0.5 [69]. Its results were parsed to the masked invariant “exposed” proteome, and only conserved fragments of at least 15 amino acids that were also predicted as potential epitopes by BepiPred-2.0 were considered.

(ii) For the structure-based approach, we used BLASTP to search the list of proteins predicted as “exposed” against the Protein Data Bank (PDB) database [49]. We only retrieved one hit, ASFV’s major capsid protein p72, with two structures (PDB IDs: 6KU9 [50] and 6L2T [9]). We used these pdb files to measure flexibility of protein residues by calculating the normalized B-value, or temperature factor, of each alpha carbon with the equation:(1)$$Zb=\frac{B-\mu b}{\sigma_{B}}$$
where *Z_B_* is the normalized B-value, *B* is the raw B-value extracted from the PDB file, *μ_B_* is the mean B-value of all alpha carbons of the structure, and *σ_B_* is the standard deviation of all the B-values of all alpha carbons [67]. Additionally, we used NACCESS [86], kindly shared by Dr. Hubbard, to calculate the RSA of each residue. We considered all linear peptide fragments of at least eight residues long with a *Z_B_* of at least 1.00 and a total mean RSA above 50% as good candidates to be potential epitopes [67]. Structurally predicted epitopes were parsed against the ASFV “exposed” invariant proteome to ensure their conservation. Epitopes were mapped on their respective PDB structure with PyMOL Molecular Graphics System (version 2.3.0, Schrödinger LLC).

### 4.8. In-House Scripts and BLAST Searches

All the work has been completed with publicly available computational tools and in-house scripts written in Python 3 [87]. The computational pipeline followed can be found in GitHub (https://github.com/ros-luc/ASFV-epitopes).

We blasted all epitopes, either those retrieved from IEDB or those predicted de novo, against protein sequences present in pigs to find out the probability of cross-reactivity. For this purpose, epitopes were first blasted against *Sus scrofa* (taxid: 9823) RefSeq protein database using BLASTP [88]. We adjusted the program to run using the PAM30 matrix, with an e-value threshold of 200,000, a word size of 2, an existence 9 and extension 1 gap cost, and without compositional adjustments [89]. Additionally, to find out epitopes’ identity to those proteins present in the pig microbiota, we performed a BLASTP search against a database containing all NCBI available RefSeq sequences of bacterial species within the reported genera from swine microbiota [90,91,92]. We were not able to include any sequences related to other microbiota components, such as bacteriophages and other viruses, fungi, or parasites, due to lack of data regarding these.

## 5. Conclusions

We identified 49 epitopes, both B-cell and T-cell (CD4^+^ and CD8^+^) epitopes, which have the potential to be used as part of an epitope-based vaccine for ASF. They are all strictly conserved among ASF viruses, and show low identity rates to proteins in *S. scrofa* and its microbiota. Still, only four of them have experimental evidence of being appropriately processed, presented and recognized by the immune system. Thus, the next step of this work will entail the experimental validation of those epitopes predicted de novo. Their functional characterization by in vitro immunological assays including, if possible, surrogates of immunogenicity and protection should also be performed. Only then may an epitope-based vaccine encompassing them be conceived and moved onto in vivo assays.

## Figures and Tables

**Figure 1 pathogens-09-01078-f001:**
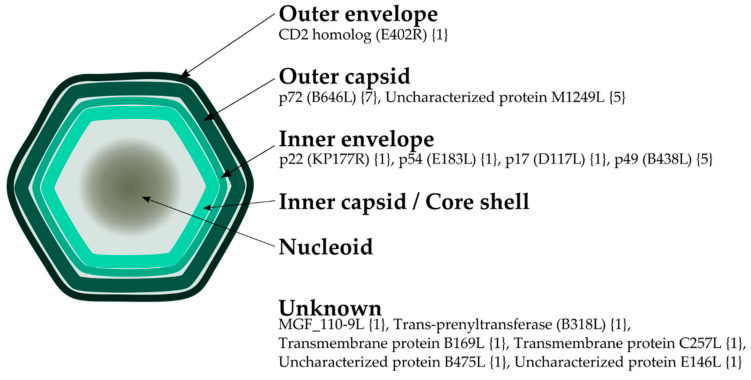
Schematic view of the viral particle morphology showing the localization of African swine fever virus (ASFV) proteins where B-cell epitopes were predicted following a sequence-based approach. Name of open reading frames (ORFs), when applicable, is indicated in parenthesis; number of predicted epitopes is shown in brackets.

**Figure 2 pathogens-09-01078-f002:**
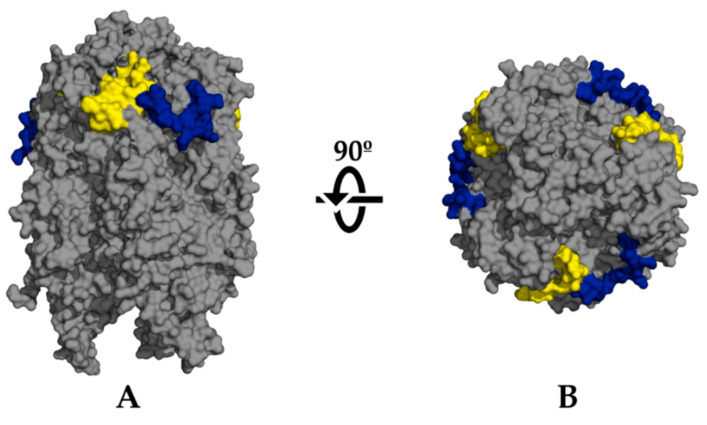
PyMOL visualization of the selected B-cell epitopes predicted following a structure-based approach in the major capsid protein p72. Epitopes KPHQSKPILTDENDTQRT (yellow) and NIQTAGKQDITPITD (blue) are mapped onto PDB ref. 6L2T. (**A**) Side view. (**B**) Top view.

**Figure 3 pathogens-09-01078-f003:**
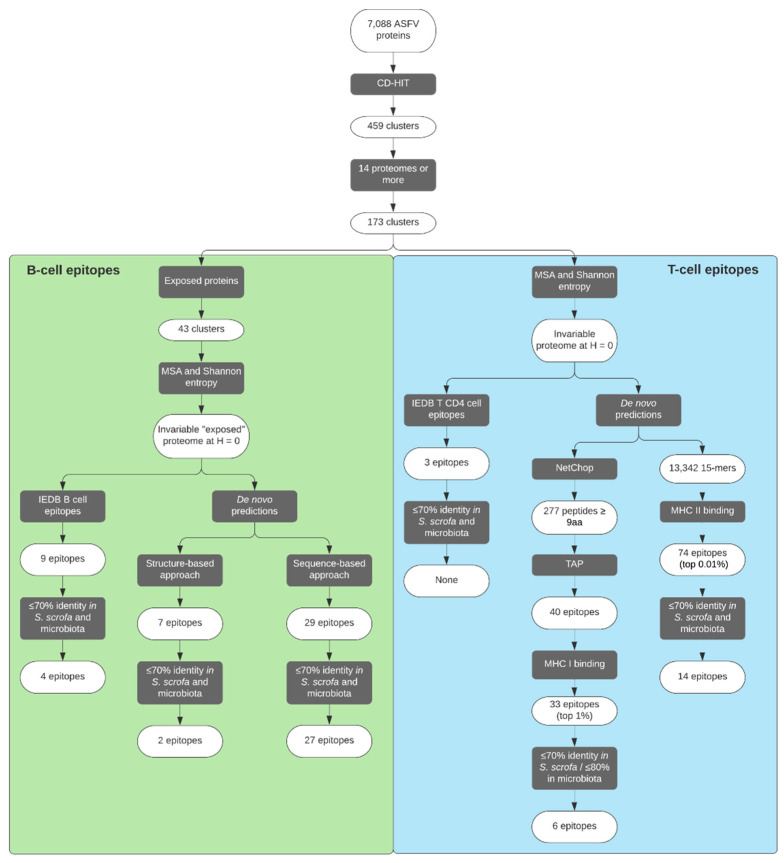
Study flowchart depicting the procedures followed to reach the selected epitopes.

**Table 1 pathogens-09-01078-t001:** ASFV conserved and experimentally validated B-cell epitopes from the IEDB.

Epitope	Protein Name (ORF)	Protein ID ^1^	*Sus scrofa* Hit (id %) ^2^	Microbiota Hit (id %) ^2^
FPENSH*NIQTAGKQD*	p72 (B646L)	E0WMM0	XP_020938976.1 (40.0%)	WP_008983921.1 (53.3%)
YCEYPGERLYENVRFDVNGNSLDEYSSDVTTL	p72 (B646L)	E0WMM0	XP_020953642.1 (25.0%)	WP_091148772.1 (34.4%)
VCKVDKDCGSGEHCV	p22 (KP177R)	E0WMB7	XP_020923036.1 (66.7%)	WP_046972048.1 (60.0%)
AAIEEEDIQFINPYQD	p54 (E183L)	E0WM75	XP_020932305.1 (50.0%)	WP_042346777.1 (68.8%)

^1^ Protein ID as listed in UniProt. ^2^ Hit reference provided as NCBI accession number; its percentage of identity to the predicted epitope sequence is provided in parenthesis. Underlined residues match with sequence-based predicted epitopes. Residues underlined and in italics match with both sequence-based and structure-based predicted epitopes.

**Table 2 pathogens-09-01078-t002:** List of conserved ASFV B-cell epitopes predicted with a sequence-based approach (continues in next page).

Epitope	Protein Name (ORF)	Protein ID ^1^	*Sus scrofa* Hit (id %) ^2^	Microbiota Hit (id %) ^2^
LATCGKAGNFCECSNYSTS	CD2 homolog (E402R)	E0WMJ6	XP_003483701.1 (47.4%)	WP_007633917.1 (47.4%)
KKQQPPKKVCKVDKDCGSGEHC	p22 (KP177R)	E0WMB7	XP_020923055.1 (40.9%)	WP_137027291.1 (63.6%)
FFQPVYPRHYGECLSP	p54 (E183L)	E0WM75	XP_020934554.1 (50.0%)	WP_018576392.1 (56.3%)
GFEYNKVRPHTGTPTLGNKLT	p72 (B646L)	E0WMM0	XP_020939243.1 (42.9%)	WP_143542922.1 (47.6%)
H*KPHQSKPILTDENDTQRT*C	p72 (B646L)	E0WMM0	XP_020932317.1 (35.0%)	WP_169883328.1 (45.0%)
HTNPKFLSQHFPENSH*NIQTAGKQDITPITD*	p72 (B646L)	E0WMM0	XP_020933726.1 (32.3%)	WP_042032899.1 (38.7%)
QMGAHGQLQTFPRNGYDWDNQTPLE	p72 (B646L)	E0WMM0	XP_020956625.1 (28.0%)	WP_075838746.1 (36.0%)
SFQDRDTALPDACSSISDI	p72 (B646L)	E0WMM0	XP_020950058.1 (42.1%)	WP_165776787.1 (57.9%)
TWNISDQNPHQHRDWHK	p72 (B646L)	E0WMM0	XP_003123199.2 (41.2%)	WP_077451423.1 (58.8%)
VTHTNNNHHDEKLMS	p72 (B646L)	E0WMM0	XP_020946598.1 (46.7%)	WP_072834091.1 (53.3%)
TSPLLSHNLSTREGIKQ	p17 (D117L)	E0WM57	XP_020947378.1 (47.1%)	WP_021704232.1 (58.8%)
GVCKNKVFEKHPLIKKNDY	MGF_110-9L	E0WMD4	XP_020922529.1 (47.4%)	WP_124535981.1 (63.2%)
AGRGIPLGNPHVKPNIEQELIKS	p49 (B438L)	E0WML4	XP_013850764.1 (30.4%)	WP_168007327.1 (56.5%)
FPKDFNASSVPLTSAEKDHSLRGDNS	p49 (B438L)	E0WML4	NP_999435.1 (50.0%)	WP_007759066.1 (46.2%)
GQAEYFDTSKQTISRHNNYIPKYTGGIGDS	p49 (B438L)	E0WML4	XP_020939832.1 (33.3%)	WP_095912160.1 (36.7%)
LADYRSDPPLWESDLPRHNRYSDNILN	p49 (B438L)	E0WML4	XP_020954566.1 (25.9%)	WP_143667676.1 (55.6%)
LNPQHKNIGYGDAQDLEPYS	p49 (B438L)	E0WML4	XP_013849660.1 (30.0%)	WP_085490987.1 (60.0%)
PSFDNDVKRRNKDTVWARFGV	Trans-prenyltransferase (B318L)	E0WML3	XP_005674076.1 (38.1%)	WP_142159516.1 (47.6%)
GINNLGEKIYTCEPFKTSF	Transmembrane protein B169L	E0WML5	XP_003354140.1 (42.1%)	WP_095898538.1 (63.2%)
KYLERQDLELLGYSPT	Transmembrane protein C257L	E0WMK4	XP_003356508.3 (68.8%)	WP_111879816.1 (62.5%)
DEPIVQNPFLENFWKPEQKTFNQSGLFEESS	Uncharacterized protein B475L	E0WML6	XP_020932571.1 (29.0%)	WP_078933170.1 (38.7%)
FRDAQNPPSSFTLGGHCQA	Uncharacterized protein E146L	E0WM78	NP_001121958.1 (52.6%)	WP_147756634.1 (47.4%)
EASYDTMRTKLMKFSGINKEKENN	Uncharacterized protein M1249L	E0WMJ8	XP_020928637.1 (41.7%)	WP_022932625.1 (41.7%)
GKQEAELITTEYLNIKKQWELQEKNACA	Uncharacterized protein M1249L	E0WMJ8	XP_020951797.1 (50.0%)	WP_134792973.1 (46.4%)
ISAYSSPGLFGEDIID	Uncharacterized protein M1249L	E0WMJ8	XP_020934753.1 (50.0%)	WP_041516260.1 (56.3%)
LSQPELVHDYYNNYKDQY	Uncharacterized protein M1249L	E0WMJ8	XP_020923164.1 (38.9%)	WP_133014505.1 (55.6%)
NPVEQKFLQHAEQREKEQMILQ	Uncharacterized protein M1249L	E0WMJ8	XP_020946579.1 (50.0%)	WP_085491731.1 (45.5%)

^1^ Protein ID as listed in UniProt. ^2^ Hit reference provided as NCBI accession number; its percentage of identity to the predicted epitope sequence is provided in parenthesis. Underlined sequences indicate shared residues with IEDB-validated epitopes; sequences in italics indicate shared residues with both IEDB-validated epitopes and epitopes predicted with a structure-based approach.

**Table 3 pathogens-09-01078-t003:** List of conserved ASFV B-cell epitopes predicted with a structure-based approach.

Epitope	Protein (ORF)	Protein ID ^1^	Mean Flex. ^2^	Mean RSA ^3^	*Sus scrofa* Hit (id %) ^4^	Microbiota Hit (id %) ^4^
KPHQSKPILTDENDTQRT	p72 (B646L)	E0WMM0	1.7	50.1	XP_020932317.1 (38.9%)	WP_169883328.1 (50.0%)
NIQTAGKQDITPITD	p72 (B646L)	E0WMM0	1.8	56.8	XP_020933726.1 (66.7%)	WP_001886209.1 (53.3%)

^1^ PDB entry 6L2T did not have a reference sequence tied to UniProt. However, both epitopes are found in Georgia 2007/1 p72 protein and UniProt ID for this entry is provided. ^2^ Values of flexibility (arbitrary units) and ^3^ RSA (%) calculated as explained in Materials and Methods. ^4^ Hit reference provided as NCBI accession number; its percentage of identity to the predicted epitope sequence is provided in parenthesis. Underlined sequences indicate shared residues with conserved and IEDB-validated epitopes from Table 1.

**Table 4 pathogens-09-01078-t004:** List of conserved ASFV predicted CD4^+^ T-cell epitopes.

Epitope	No. HLA Alleles ^1^	Protein Name (ORF)	Protein ID ^2^	*Sus scrofa* Hit (id %) ^3^	Microbiota Hit (id %) ^3^
GLGFILIVIFIYLLLITLQQMLTRHI	5	Uncharacterized protein (B117L)	E0WM34	XP_003360623.1 (46.2%)	WP_119317083.1 (53.9%)
MNIYLVWFLYILLGNLILAVIY	4	ASFV_G_ACD_01990	E0WMB3	XP_003124218.1 (59.1%)	WP_155703379.1 (68.2%)
LSLICVFSHFFEELYITKP	3	Probable methyltransferase (EP424R)	E0WMJ3	XP_013843985.2 (42.1%)	WP_010770249.1 (52.6%)
WYLKYVIAYILLLTMLVIGLIYRIIVLIYRSIQAQK	3	ASFV_G_ACD_01940	E0WMA9	XP_013836670.2 (33.3%)	WP_100215420.1 (55.6%)
AINFLLLQNGSAVLRYS	2	p72 (B646L)	E0WMM0	NP_999474.1 (52.9%)	WP_113638931.1 (64.7%)
ATRLVAVRAQQLAINGSTMLKKK	2	DNA-directed RNA polymerase subunit 6 homolog (C147L)	E0WMK7	XP_020956260.1 (30.4%)	WP_093129025.1 (56.5%)
GLNFQAVRYEMIMSLPLDIP	2	Putative ATP-dependent RNA helicase (D1133L)	E0WM56	XP_013834167.2 (55.0%)	WP_096721649.1 (60%)
HYPASFHYTMLEALIIDN	2	Putative ATP-dependent RNA helicase (D1133L)	E0WM56	XP_005666131.2 (38.9%)	WP_167872395.1 (55.6%)
ISIITFLSLRKRKKHVEEI	2	CD2 homolog (E402R)	E0WMJ6	XP_020951930.1 (42.1%)	WP_039931062.1 (52.6%)
MYFQQTRSILIKNDAVFILNLG	2	ASFV_G_ACD_01870	E0WMA2	XP_020949404.1 (54.6%)	WP_069997539.1 (40.9%)
NAFVDYIISNFNHAVTCRKP	2	Transmembrane protein (B66L)	E0WM38	XP_020936233.1 (40.0%)	WP_128359657.1 (45.0%)
NNILVEILSFKNYYSSNTSLLSIKT	2	MGF_360-21R	E0WMB1	XP_020924396.1 (40.0%)	WP_117888406.1 (44.0%)
NVIFLKVISNTAVSVFWRD	2	Uncharacterized protein (C62L)	E0WMK8	XP_020958320.1 (52.6%)	WP_094566288.1 (52.6%)
PTPLIPSMAMSIPRMINKRKKRIQFLTFLTNLFLYN	2	Uncharacterized protein (A118R)	E0WMH4	XP_020945397.1 (30.6%)	WP_165296256.1 (50.0%)

^1^ All predicted epitopes presented a binding affinity to HLA class II alleles within the top 0.1% score. Specific affinities for each allele are provided in Appendix A. ^2^ Protein ID as listed in UniProt. ^3^ Hit reference provided as NCBI accession number; its percentage of identity to the predicted epitope sequence is provided in parenthesis.

**Table 5 pathogens-09-01078-t005:** List of conserved ASFV-predicted CD8^+^ T-cell epitopes.

Predicted Epitope	No. SLA Alleles ^1^	Protein (ORF)	Protein ID ^2^	TAP IC_50_ ^3^	*Sus scrofa* Hit (id %) ^4^	Microbiota Hit (id %)^4^
MAMQKLFTY	29	Putative DNA-directed RNA polymerase subunit 5 homolog (D205R)	E0WM58	−0.94	NP_001037992.1 (66.7%)	WP_025027576.1 (77.8%)
KRHENIWML	27	Uncharacterized protein D339L	E0WM55	−2.81	XP_013850843.2 (55.6%)	WP_130548297.1 (77.8%)
CTQPARVTY	22	DNA-directed RNA polymerase subunit beta (EP1242L)	E0WMJ1	0.17	XP_003127876.2 (66.7%)	WP_164721234.1 (77.8%)
NIMPGLVSY	21	Ribonucleoside-diphosphate reductase (F334L)	E0WMI2	0.65	XP_020927225.1 (55.6%)	WP_157084930.1 (77.8%)
ANPSEGWKY	10	DNA topoisomerase 2 (P1192R)	E0WM62	0.23	XP_020958373.1 (55.6%)	WP_117029589.1 (77.8%)
EEFNYLWVY	7	Uncharacterized protein G1340L	E0WM39	−2.05	NP_999483.1 (66.7%)	WP_083089560.1 (77.8%)

^1^ Number of different SLA class I alleles bound by the epitopes predicted within the top 1% score. Specific affinities for each allele are provided in Appendix A. ^2^ Protein ID as listed in UniProt. ^3^ Binding values given as log(IC50). ^4^ Hit reference provided as NCBI accession number; its percentage of identity to the predicted epitope sequence is provided in parenthesis.

## Data Availability

The data presented in this study is contained within the article or supplementary material. Besides, the scripts used are openly available in https://github.com/ros-luc/ASFV-epitopes/search?q=ASFV-epitopes.

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
