# Peer review of "Computational Analysis of African Swine Fever Virus Protein Space for the Design of an Epitope-Based Vaccine Ensemble"

_pathogens, 2020, doi:10.3390/pathogens9121078_

Round 1

Reviewer 1 Report

Albert Ros-Lucas and co-authors describe the bioinformatic approach to identify epitopes of African swine fever virus (ASFV) that may be recognised by the host immune system and therefore could be potential targets for development of epitope-based vaccine.

I would like to congratulate the authors on the clear and logical presentation of methods and results. Both were described in enough detail for a competent bioinformatician to reproduce the analyses (including the link to a computational pipeline in GitHub), but also in a plain enough language to allow scientists with more basic bioinformatic knowledge to follow the methodology.

The authors used a variety of computational tools to predict B-cell and T-cell epitopes that are conserved between various ASF viruses circulating in the field. They identified 29 B-cell, 14 CD4+ T-cell and 6 CD8+ T-cell epitopes as an outcome of their analyses. Further experimental studies are needed to confirm whether or not these epitopes evoke protective immunological responses, which is stated by the authors.

Overall, the manuscript is well written, and the scientific approach used seems valid. A have only a few comments for the authors’ consideration:

Methods: The authors considered epitopes that were specific for ASFV and had low similarity (and hence low probability of cross-reaction) with host proteins and with proteins from bacteria predicted to be present in the porcine gut (“pig microbiota”). I have two comments regarding this approach:

  • Could the authors elaborate a little bit further on why they excluded epitopes that may be cross-reactive with the gut microbiota of pigs? The explanation provided in lines 282-286 seems to relate to systemic IgG antibodies and in fact, everywhere in the manuscript the authors mention protection offered by ASFV-specific IgG antibodies (e.g. line 295). How would cross-reaction with epitopes from gut microbiota affect the efficacy of a potential parenteral vaccine? Or is the planned vaccination strategy to elicit local IgA antibodies within the alimentary tract? The authors briefly mention in the introduction that gut microbiota may be important for the defence against ASFV infection (line 92), but it is not clear how this relates to the methods and results presented in the manuscript.
  • Also, I wonder why only bacterial epitopes were considered for the assessment of potential cross-reaction with gut microbiota. Wouldn’t it be expected that a large variety of bacteriophages are also present in that environment?

Line 226: any epitopes (plural)

Lines 256-258: The information presented in these two sentences seems to be contradictory. First, the authors state that “immunization with the low virulent strain NHV/1968 provided protection against the closely related but highly virulent strain L60”, which is followed by the seemingly contradictory statement that “protection was thus only attained against a homologous challenge”. Could this please be clarified?

Line 258-263: The authors stated that the CD-2 deficient virus “offered protection against the parental strain, and also against challenge with heterologous strains E75 and Georgia 2007/1”. However, in the next sentence it is stated that “the same principle of generating a CD2 homolog-deficient virus did not work for the Georgia 2007/1 strain”. Could the authors possibly re-word this section to highlight the fact that conflicting results were reported by different groups and maybe offer some explanation for those discrepancies?

Line 304: I suggest “This protein is one of the most divergent among different ASF viruses” or “The sequence coding for this protein is one of the most divergent between ASFV genomes” instead of the current version.

Line 305 and elsewhere: “between different viruses” would be a more accurate description that the current version (“between different strains”). A strain is traditionally a virus that has been propagated in the lab and has stable well-characterised properties (e.g. a vaccine strain). Hence, while all the strains are viruses, not every virus or viral isolate is a strain. The authors are encouraged to refer to a publication by Kuhn and others (https://www.ncbi.nlm.nih.gov/pmc/articles/PMC2878132/ ), in which the proper use of viral nomenclature is explained (including the use of the word “strain”).

Line 313: I suggest “…both epitopes identified from the analysis of p72 3D structures…” instead of the current version. Also, a reference to Figure 2 would fit better here than a reference to Table 3.

Line 314: delete two “with” e.g. “match with sequences” and “and with experimentally validated sequences”

Line 317: “Wang et al described” (past tense)

Line 318: I suggest “…and it may be targeted by antibodies in a “naked” particle” instead of the current version.

Line 326: “within inner layers”

Line 338: “as it was the case with”

Line 343: “On the other hand” or just delete this altogether and start the sentence with “Previous experiments…”

Line 434: Could the authors insert a reference to support the necessity of the TM domain to be present for the protein to be membrane-associated with surface-exposed epitopes?

Author Response

Comments and Suggestions for Authors: Albert Ros-Lucas and co-authors describe the bioinformatic approach to identify epitopes of African swine fever virus (ASFV) that may be recognised by the host immune system and therefore could be potential targets for development of epitope-based vaccine.

I would like to congratulate the authors on the clear and logical presentation of methods and results. Both were described in enough detail for a competent bioinformatician to reproduce the analyses (including the link to a computational pipeline in GitHub), but also in a plain enough language to allow scientists with more basic bioinformatic knowledge to follow the methodology.

The authors used a variety of computational tools to predict B-cell and T-cell epitopes that are conserved between various ASF viruses circulating in the field. They identified 29 B-cell, 14 CD4+ T-cell and 6 CD8+ T-cell epitopes as an outcome of their analyses. Further experimental studies are needed to confirm whether or not these epitopes evoke protective immunological responses, which is stated by the authors.

Overall, the manuscript is well written, and the scientific approach used seems valid. A have only a few comments for the authors’ consideration:

Methods: The authors considered epitopes that were specific for ASFV and had low similarity (and hence low probability of cross-reaction) with host proteins and with proteins from bacteria predicted to be present in the porcine gut (“pig microbiota”). I have two comments regarding this approach:

- Could the authors elaborate a little bit further on why they excluded epitopes that may be cross-reactive with the gut microbiota of pigs? The explanation provided in lines 282-286 seems to relate to systemic IgG antibodies and in fact, everywhere in the manuscript the authors mention protection offered by ASFV-specific IgG antibodies (e.g. line 295). How would cross-reaction with epitopes from gut microbiota affect the efficacy of a potential parenteral vaccine? Or is the planned vaccination strategy to elicit local IgA antibodies within the alimentary tract? The authors briefly mention in the introduction that gut microbiota may be important for the defence against ASFV infection (line 92), but it is not clear how this relates to the methods and results presented in the manuscript.

We thank the reviewer for his/her comment, and have modified the manuscript accordingly. We hope that the explanation provided in response to his/her question will help to justify why we think that developing an immune response against self-microbiota should be avoided. Optimal vaccines should be able not only to induce systemic immune responses, but also mucosal immunity, including in the oral-nasal cavities, capable of blocking the pathogens at the entry site. However, it is crucial to avoid the destruction of beneficial microbiota in such cavities by inducing undesired immune responses against it. The role of microbiota in body homeostasis, immune system maturation and pathogen defense has been demonstrated (https://doi.org/10.3389/fimmu.2019.00607), therefore recommending to avoid any uncontrolled alteration of its composition. Furthermore, the recent confirmation of gut microbiota as a relevant player in ASF resistance (https://doi.org/10.1038/s41598-020-74651-3; https://doi.org/10.1128/JVI.01893-14), reinforces our idea of avoiding undesired vaccine-induced immunity against microbiota. On top of that, it has been reported that epitopes conserved in microbiota show inferior levels of immunogenicity due to the tolerance of the immune system to the commensal bacteria (https://doi.org/10.1145/2649387.2660843), which supports the idea of discarding these epitopes.

- Also, I wonder why only bacterial epitopes were considered for the assessment of potential cross-reaction with gut microbiota. Wouldn’t it be expected that a large variety of bacteriophages are also present in that environment? Unfortunately, most of the microbiota data available for swine refers only to bacteria, complicating a careful analysis of the rest of the microbiota components, including not only bacteriophages, but also other viruses, fungi, parasites... We have now reflected this limitation in the materials and methods section.

Line 226: any epitopes (plural)
Plural has now been used.

Lines 256-258: The information presented in these two sentences seems to be contradictory. First, the authors state that “immunization with the low virulent strain NHV/1968 provided protection against the closely related but highly virulent strain L60”, which is followed by the seemingly contradictory statement that “protection was thus only attained against a homologous challenge”. Could this please be clarified?
We thank again the reviewer for his/her observation. We have re-worded this fragment to clarify this apparent confusion. Comparison of the full-length genomes of L60 and NHV1968 allowed hypothesizing that the latter was the result of intensive passage of the L60 (or an L60-like) virulent virus in tissue culture during the early 1960s to obtain an attenuated vaccine later used in the field (https://doi.org/10.1099/vir.0.070508-0), therefore being considered homologous and explaining the experimental protection observed. Anyhow, the fine line existing between homologous and heterologous concepts recommends being cautious with their use in the future.

Line 258-263: The authors stated that the CD-2 deficient virus “offered protection against the parental strain, and also against challenge with heterologous strains E75 and Georgia 2007/1”. However, in the next sentence it is stated that “the same principle of generating a CD2 homolog-deficient virus did not work for the Georgia 2007/1 strain”. Could the authors possibly re-word this section to highlight the fact that conflicting results were reported by different groups and maybe offer some explanation for those discrepancies?

Many reasons might explain this differential phenotype, including the existence of redundant ORFs in different ASF viruses encoding proteins capable of covering the absence of some essential function played by CD2v. This is not unique for CD2v, since it has been also described for other viral factors, such as NL gene, UK gene and 9GL (https://doi.org/10.1038/s41598-020-57455-3; https://doi.org/10.1128/jvi.01760-16). In this regard, Georgia 2007/1 appears to be much more complex than Ba71, the former having more than 30 additional hypothetical proteins that the latter, which might involve supplemental functions as mentioned above. This fact further reinforces our idea of focusing our analysis on the current circulating virus in Eurasia, in order to tackle this complexity as much as possible. We have made changes in the text to clarify these discrepancies.

Line 304: I suggest “This protein is one of the most divergent among different ASF viruses” or “The sequence coding for this protein is one of the most divergent between ASFV genomes” instead of the current version.
The sentence has been corrected.

Line 305 and elsewhere: “between different viruses” would be a more accurate description that the current version (“between different strains”). A strain is traditionally a virus that has been propagated in the lab and has stable well-characterised properties (e.g. a vaccine strain). Hence, while all the strains are viruses, not every virus or viral isolate is a strain. The authors are encouraged to refer to a publication by Kuhn and others (https://www.ncbi.nlm.nih.gov/pmc/articles/PMC2878132/ ), in which the proper use of viral nomenclature is explained (including the use of the word “strain”).
We have corrected the text accordingly.

Line 313: I suggest “…both epitopes identified from the analysis of p72 3D structures…” instead of the current version. Also, a reference to Figure 2 would fit better here than a reference to Table 3.
This has been corrected as suggested.

Line 314: delete two “with” e.g. “match with sequences” and “and with experimentally validated sequences”
This has been corrected.

Line 317: “Wang et al described” (past tense)
Past tense has been used.

Line 318: I suggest “…and it may be targeted by antibodies in a “naked” particle” instead of the current version. The sentence has been corrected.

Line 326: “within inner layers”
Text has been corrected.

Line 338: “as it was the case with”
Text has been corrected.

Line 343: “On the other hand” or just delete this altogether and start the sentence with “Previous experiments…”
Text has been corrected.

Line 434: Could the authors insert a reference to support the necessity of the TM domain to be present for the protein to be membrane-associated with surface-exposed epitopes?
When tackling B-cell epitopes, we were limited by the fact that only CD2 has been described as exposed in the outer lipid layer of the virus. Since experimental data suggests the existence of neutralizing antibodies against proteins such as p72 or p54, we wanted to expand our search. We used the recent work by Wang et al. (https://doi.org/10.1126/science.aaz1439) to select the proteins of the capsid. However, to tackle other putative exposed proteins located in the lipid membranes, we had to rely on predictions. We considered as the most accurate criteria the presence of transmembrane domains as predicted by TMHMM, which would indeed indicate binding to lipid membranes. This way, we could also discard proteins that might be exposed (i.e. soluble proteins), but that might be viral factors distracting the immune system. Upon the epitope prediction by BepiPred, we mapped all predicted epitopes onto their proteins of origin and compared their positioning to what TMHMM predicted as "outside", "inside" or "transmembrane" regions. Any epitopes that fell into the later two categories were discarded, this way trying to make sure, as much as we could, that epitopes were indeed putatively exposed. We changed the text to reflect that while having a transmembrane domain gives a higher change of being exposed, it does not ensure this feature.

Reviewer 2 Report

The authors present a predictive in silico approach to identify potential ASFV immunogenic epitopes that can be used towards ASFV vaccine development. The authors give a good review of the background literature on ASFV vaccine approaches and viral antigens, discuss the gaps in knowledge and rational for this study. The results and methods are clearly written and easy to follow. The figures and tables are appropriate and informative. While there is the obvious limitation of the results being only predictive and the analysis unable to predict conformational epitopes, the authors acknowledge these limitations. Presented is a refined list of a reasonable number of potential candidates to pursue and refine further in future in vitro and in vivo studies. The manuscript reads well and gives a good discussion of the findings which are of interest to the ASF research community and the readers of Pathogens.

Author Response

Comments and Suggestions for Authors: The authors present a predictive in silico approach to identify potential ASFV immunogenic epitopes that can be used towards ASFV vaccine development. The authors give a good review of the background literature on ASFV vaccine approaches and viral antigens, discuss the gaps in knowledge and rational for this study. The results and methods are clearly written and easy to follow. The figures and tables are appropriate and informative. While there is the obvious limitation of the results being only predictive and the analysis unable to predict conformational epitopes, the authors acknowledge these limitations. Presented is a refined list of a reasonable number of potential candidates to pursue and refine further in future in vitro and in vivo studies. The manuscript reads well and gives a good discussion of the findings which are of interest to the ASF research community and the readers of Pathogens.

Thank you very much.

Reviewer 3 Report

In this article Ros-Lucas et al., analyse ASFV sequences by immunoinformatic tools, to identify potential B- and T-cell epitopes relevant for vaccines design. The search of B-cell epitopes was done on exposed proteins to identify neutralizing epitopes. For that purpose two approaches were followed; one based on sequence analyses and other on structure. Furthermore, T-cell epitopes CD4+ and CD8+ were also predicted.

African swine fever is a quite relevant disease that affect wild and domestic pigs causing significant economic concerns to swine industry. Currently, the virus causing this disease the ASFV is circulating in Europe and Asia causing frequent outbreaks. The virus strains circulating derived from the 2007 Georgia isolate, a genotype II virus. There is no available vaccine against ASF so the control disease relies on the culling of infected animals.

In this context the aim of the present article is quite relevant. The identification of potential targets for the development of effective vaccines is an important goal. In this paper the authors address this objective using an in-silico approach, selecting adequate criteria for epitope selection as well as the immunoinformatic tools more adequate to this end. The innovative part of this study is the systematic analyses of exposed proteins,conserved among circulating strains that allow identification of novel epitopes additional to those already included in the Immune Epitope Database.

However, the paper have certain weaknesses which limit the interest of the results:

- It would be more valuable to include in this article any test to demonstrate at least the antigenicity of the selected epitopes. In the case of B-cell epitopes maybe using sera from a collection already available at lab is affordable to evaluate by ELISA if sera from ASFV infected pigs recognize or not such sequences. Why didn’t the authors do this kind of analyses?

- In regard of T-cell epitopes is crucial to know the SLA binding affinity of CD4+ and CD8+ epitopes selected, to the SLA alleles more frequently represented in outbred pig populations. To this end it would be interesting to include an analyses in-silico of the main SLA alleles that can recognise the T-cell epitopes selected and a comment in the discussion about the potential population coverage of such epitopes for European farmed pigs (probably less variable than those from Asia).

Author Response

Comments and Suggestions for Authors: In this article Ros-Lucas et al., analyse ASFV sequences by immunoinformatic tools, to identify potential B- and T-cell epitopes relevant for vaccines design. The search of B-cell epitopes was done on exposed proteins to identify neutralizing epitopes. For that purpose two approaches were followed; one based on sequence analyses and other on structure. Furthermore, T-cell epitopes CD4+ and CD8+ were also predicted.

African swine fever is a quite relevant disease that affect wild and domestic pigs causing significant economic concerns to swine industry. Currently, the virus causing this disease the ASFV is circulating in Europe and Asia causing frequent outbreaks. The virus strains circulating derived from the 2007 Georgia isolate, a genotype II virus. There is no available vaccine against ASF so the control disease relies on the culling of infected animals.

In this context the aim of the present article is quite relevant. The identification of potential targets for the development of effective vaccines is an important goal. In this paper the authors address this objective using an in-silico approach, selecting adequate criteria for epitope selection as well as the immunoinformatic tools more adequate to this end. The innovative part of this study is the systematic analyses of exposed proteins, conserved among circulating strains that allow identification of novel epitopes additional to those already included in the Immune Epitope Database.

However, the paper have certain weaknesses which limit the interest of the results:

- It would be more valuable to include in this article any test to demonstrate at least the antigenicity of the selected epitopes. In the case of B-cell epitopes maybe using sera from a collection already available at lab is affordable to evaluate by ELISA if sera from ASFV infected pigs recognize or not such sequences. Why didn’t the authors do this kind of analyses? This would, in fact, be the ideal scenario, and the limitation is acknowledged in the text. Georgia 2007/1 causes such a severe disease that pigs do not survive long enough to develop an immune response, and since there is no commercial vaccine available, the only way to obtain sera would be by using experimental live-attenuated vaccines in high-security BSL-3 labs. With the eventual publication of this manuscript, we hope to be able to secure funding so that we can continue with this line of experimentation performing all required wet lab experiments.

- In regard of T-cell epitopes is crucial to know the SLA binding affinity of CD4+ and CD8+ epitopes selected, to the SLA alleles more frequently represented in outbred pig populations. To this end it would be interesting to include an analyses in-silico of the main SLA alleles that can recognise the T-cell epitopes selected and a comment in the discussion about the potential population coverage of such epitopes for European farmed pigs (probably less variable than those from Asia). In our search for T-cell epitopes, we prioritized those with a wide coverage of MHC alleles. The suggested analysis cannot be made for CD4 epitopes with SLA class II since there is currently no tool available trained and suited for predicting class II epitopes in the pig immunological context. On the other hand, we selected those CD8 epitopes that were predicted to bind to a wide variety of SLA class I alleles (see Table 5). For example, epitopes MAMQKLFTY and KRHENIWML were respectively predicted to bind to 29 and 27 different SLA class I alleles (Supplementary Table 4), which would ensure wide population coverage. In total, the 6 CD8 T-cell epitopes finally selected would cover 38 out of the 45 SLA class I alleles eligible in the prediction tool we used. Text has been reworded to reflect this.